# Detecting the Neuraminidase R294K Mutation in Avian Influenza A (H7N9) Virus Using Reverse Transcription Droplet Digital PCR Method

**DOI:** 10.3390/v15040983

**Published:** 2023-04-17

**Authors:** Xiuyu Lou, Hao Yan, Lingxuan Su, Yi Sun, Xinyin Wang, Liming Gong, Yin Chen, Zhen Li, Zhongbiao Fang, Haiyan Mao, Keda Chen, Yanjun Zhang

**Affiliations:** 1Key Laboratory of Public Health Detection and Etiological Research of Zhejiang Province, Department of Microbiology, Zhejiang Provincial Center for Disease Control and Prevention, Hangzhou 310051, China; xylou@cdc.zj.cn (X.L.);; 2Shulan International Medical College, Zhejiang Shuren University, Hangzhou 310015, China

**Keywords:** absolute quantitation, avian influenza virus H7N9, R294K mutation, RT-droplet digital PCR

## Abstract

The R294K mutation in neuraminidase (NA) causes resistance to oseltamivir in the avian influenza virus H7N9. Reverse transcription droplet digital polymerase chain reaction (RT-dd PCR) is a novel technique for detecting single-nucleotide polymorphisms. This study aimed to develop an RT-dd PCR method for detecting the R294K mutation in H7N9. Primers and dual probes were designed using the H7N9 NA gene and the annealing temperature was optimized at 58.0 °C. The sensitivity of our RT-dd PCR method was not significantly different from that of RT-qPCR (*p* = 0.625), but it could specifically detect R294 and 294K in H7N9. Among 89 clinical samples, 2 showed the R294K mutation. These two strains were evaluated using a neuraminidase inhibition test, which revealed that their sensitivity to oseltamivir was greatly reduced. The sensitivity and specificity of RT-dd PCR were similar to those of RT-qPCR and its accuracy was comparable to that of NGS. The RT-dd PCR method had the advantages of absolute quantitation, eliminating the need for a calibration standard curve, and being simpler in both experimental operation and result interpretation than NGS. Therefore, this RT-dd PCR method can be used to quantitatively detect the R294K mutation in H7N9.

## 1. Introduction

In 2013, cases of human infection with H7N9, a subtype of the highly pathogenic avian influenza virus, were first reported in Shanghai and Anhui. Since then, it has become a global public health concern [1]. Five waves of H7N9 outbreaks have spread to most areas in China since 2013 and have resulted in a fatality rate of 40% [2]. The major symptoms of H7N9 infection are pneumonia and acute respiratory distress [3]. Due to the potential severity of illness associated with H7N9 virus infection, the Chinese Center for Disease Control and Prevention recommends oseltamivir treatment within 48 h of disease onset [4]. Additionally, the Centers for Disease Control and Prevention in the USA recommend treating all confirmed or probable cases of H7N9 infection with neuraminidase inhibitors (NAIs) as early as possible [5]. NAIs are specific anti-influenza virus drugs that target neuraminidase (NA), such as oseltamivir, peramivir and zanamivir [3], making them very important treatment options for patients with H7N9.

However, reports have emerged of influenza viruses that are resistant to NAI drugs. In 2013, a human H7N9 virus isolate (A/shanghai/1/2013) was identified as being resistant to oseltamivir [1]. This drug resistance is linked to mutations in the NA protein, which plays a crucial role in releasing mature virions from the host cell surface by hydrolyzing sialic acid. As such, mutations in the NA gene and misuse of NAIs can result in drug-resistant virus strains. Of 14 H7N9 patients treated with oseltamivir, 2 were found to have developed the R294K mutation [6]. This suggests that there may be a low barrier for developing drug resistance among H7N9 patients treated with oseltamivir. Wu et al. hypothesized that the A/Shanghai/1/2013 strain must have arisen from oseltamivir treatment [4]. Mutations in R294K, E119V and I222K located in NA are known to reduce the effectiveness of oseltamivir. The R294K mutation is also associated with poor clinical outcomes and can even lead to death [6,7]. The emergence of drug-resistant viruses poses a threat to clinical therapy and public health. Monitoring H7N9 virus resistance is, therefore, crucial for disease control and treatment.

Currently, the main approaches employed to detect influenza virus resistance are phenotypic and genotypic analyses. Early detection is crucial for effective drug selection and disease control. However, these methods may lack the necessary sensitivity or be slow in detecting the virus. For instance, phenotyping demands a significant amount of time and laboratory resources, whereas genotyping might not identify low-frequency mutations [8]. Hence, there is a need for a more efficient and sensitive method to detect resistance to the influenza virus. A novel technology called reverse transcription droplet digital polymerase chain reaction (RT-dd PCR) has recently emerged as an absolute quantification method for nucleic acids. Compared to other methods, RT-dd PCR is known for its high sensitivity, accuracy, and low-abundance detection, making it a popular choice in various fields, such as cancer gene and virus detection [9]. As a result, RT-dd PCR has gained widespread popularity for quantifying genomic variations, including single-nucleotide polymorphisms (SNPs) [10]. Moreover, this allows for the monitoring of changes in drug resistance genes over time, facilitating the discovery of the therapeutic effects of oseltamivir on the H7N9 virus.

The purpose of this study is to develop a highly efficient RT-dd PCR-based method for detecting drug resistance in H7N9, which will aid clinicians in selecting better drug regimens and provide essential information for controlling and treating the disease. RT-dd PCR can detect targets with very low copy numbers and quantify gene copies. Therefore, we established an RT-dd PCR platform to identify mutations in the H7N9 virus drug-resistance gene at the R294K site in NA gene. This platform can directly test clinical specimens for oseltamivir resistance. It has several advantages, such as fast analysis, simplicity, and user-friendliness. It also provides direct results for R294K-oseltamivir resistance in H7N9 viruses. Depending on the platform, the change in drug-resistance genes could be monitored over time to discover the curative effect of NAIs, such as oseltamivir, against the H7N9 virus.

## 2. Materials and Methods

### 2.1. Samples

Eighty-nine H7N9 virus case samples were provided by the laboratory of Zhejiang Provincial Center for Disease Control and Prevention (ZJCDC). These case samples were obtained during the clinical management of patients with the H7N9 virus. All case samples were cultured successfully using chicken embryos in the BSL-3 of ZJCDC.

RNA of the H1N1, H3N2, B/Victoria, B/Yamagata, H5N1, H9N2, and H7N9 virus was provided by ZJCDC. The strains used in the NAI test, GD003 R294 and GD003 R294K, were provided by the China National Influenza Center.

### 2.2. R294K Primer Design

The NA sequences of the H7N9 virus were obtained from the NCBI and GISAID influenza virus sequence databases. These sequences were aligned using Mega v.5.0 and Bioedit v.7.0 to identify the fragment within R294 and 294K. Primer Express v.3.0 was used to design the primers. Two TaqMan minor groove binder (MGB) probes (Table 1) were used, one to target the wild-type sequence and one to target the mutated sequence. All primers and probes were prepared at a concentration of 20 μmol/L, and the two primers and probes were prepared in working solutions with a volume of 2:2:1:1.

### 2.3. Plasmid Construction and Nucleic Acid Extraction

Recombinant plasmids of the wild type (WT) and mutant type (MT) of the NA gene were synthesized by Shanghai Sangon Company (Shanghai, China), which we used directly without extracting nucleic acid. When we received the plasmids, we divided the plasmids to ensure the same concentration of plasmids were used in the experiment.

H7N9 virus RNA was extracted from 200 μL of the clinical samples, including swabs, sputum and alveolar lavage fluid, using the Qiagen Rneasy Mini Kit (74104; Qiagen, Hilden, Germany), according to the manufacturer’s instructions. RNA was eluted with 50 μL of Dnase- and Rnase-free water and stored at −80 °C until use.

### 2.4. Establishing the RT-dd PCR to Detect the R294K Mutation in H7N9 Virus

#### 2.4.1. Droplet Digital PCR Detection of Plasmids and H7N9 Virus RNA

RT-dd PCR was performed using a QX200 droplet digital PCR system (Bio-Rad, Hercules, CA, USA). The dd PCR^TM^ Supermix for Probe (no dUTP) (1863024; Bio-Rad, Hercules, CA, USA) was used to amplify the plasmids. The volume of the PCR mixture was 20 μL, containing 1 μL of DNA template, 10 μL of supermix, 8 μL of ddH_2_O, and 1 μL of R294K primer/probe mix. The amplification cycling conditions included enzyme activation at 95 °C for 10 min, 40 cycles of denaturation at 94 °C for 0.5 min, annealing and extending at 58 °C for 1 min, and enzyme deactivation at 98 °C for 10 min. RNA extracted from the H7N9 virus case samples was amplified using the One-step RT-PCR Advanced Kit for Probe (1864021; Bio-Rad, Hercules, CA, USA). The reaction mixture volume was 20 μL, including 1 μL of RNA, 5 μL of supermix, 2 μL of reverse transcriptase, 1 μL of 300 mM DTT, 10 μL of ddH_2_O, and 1 μL of primer/probe mix. Amplification included the following cycling conditions: reverse transcription at 45 °C for 60 min; enzyme activation at 95 °C for 10 min, 40 cycles of denaturation at 95 °C for 0.5 min, annealing and extension at 58 °C for 1 min, and enzyme deactivation at 98 °C for 10 min. A reader was used to measure the signals of the PCR products. The total number of droplets in each reaction tube was ≥10,000, and the detection results were considered to be valid. If the number of positive droplets was ≤3, the result was determined to be negative; if the number of positive droplets was >3, it was determined to be positive. The QuantSoft v.1.7.4 software (Bio-Rad, Hercules, CA, USA) was used to determine the copy number. The results were recorded as the copy/reaction or copy/μL (copy/reaction in 20 μL of reaction).

#### 2.4.2. Determining the Optimal Annealing Temperature

The annealing temperatures were set at 60.0, 59.6, 59.0, 58.0, 56.9, 56.0, 55.3 and 55.0 °C. The template at 55.0 °C was nuclease-free water as the negative control and the same concentration of plasmid nucleic acid was used as the template for other temperatures. Two groups of parallel experiments were repeated to evaluate the degree of signal differentiation and number of nucleic acid copies, and to determine the optimal annealing temperature.

#### 2.4.3. Evaluating the Repeatability, Sensitivity, and Specificity

A ten-fold gradient was used to dilute the WT and MT plasmids three times, with the original concentration of 8146.25 copies/μL and 532 copies/μL for WT and MT, respectively, dd PCR was used to detect each dilution, and each dilution plasmid was detected in eight repeats. The mean, standard deviation (SD), and coefficient of variation (CV) were calculated to evaluate the repeatability of the dd PCR. According to the Clinical and Laboratory Standard Institute EP17-42 standard, the sensitivity of dd PCR and qPCR was evaluated using the limit of detection (LOD) of the plasmids. The reagents, reaction systems and reaction procedures used were consistent to consider the comparability of the results of dd PCR and qPCR, and the qPCR was performed on the ABI 7500. Serially dilute the plasmids 6 times at 2-fold dilution, each dilution was detected in 16 repeats using dd PCR and qPCR. The initial concentration of the mixed sample contained both MT plasmid and WT plasmid at a ratio of 1:19 with a total concentration of 10,000 copies/μL. Specifically, the WT plasmid accounted for 95% of the mixture, while the MT plasmid accounted for 5%. Further ratios were tested with the WT plasmid accounting for 97.5%, and MT plasmid for 2.5%; WT plasmid accounting for 98.75%, and MT plasmid for 1.25%; WT plasmid accounting for 99.375%, and MT plasmid for 0.625%; and finally, the WT plasmid accounting for 99.6875%, and MT plasmid for 0.3125%). Probit regression analysis was used to calculate the 95% confidence interval (CI) of the positive rate to determine the LOD. We used the RNA of influenza A (H1N1 and H3N2) virus, influenza B (Victoria and Yamagata) virus, avian influenza A (H5N1 and H9N2) virus, H7N9 virus (GD003 R294 and GD003 R294K), and nuclease-free water, as the negative control, to evaluate specificity.

#### 2.4.4. Evaluating the Detection Limit of R294K Mutation

The WT and MT plasmids were mixed, which resulted in a sequential reduction in the final MT plasmid content to 5, 2.5, 1.25, 0.625, and 0.3125%. The MT plasmid was detected using dd PCR, and each plasmid mixture was detected in eight repeats. The positivity rate was calculated to assess the effect of low-frequency detection.

### 2.5. Application of RT-dd PCR to Detect H7N9 Virus

RT-dd PCR was used to detect the H7N9 virus in the 89 samples. If the R294K mutation was detected, the respective samples were subjected to next-generation sequencing (NGS) to compare the mutation abundance.

### 2.6. R294K Mutation in Genotypic and Phenotypic Analyses

Extracted H7N9 virus RNA was reverse-transcribed to cDNA and amplified using a PathAmpFluA PCR kit (6319856; Applied Biosystems, San Francisco, CA, USA) according to the manufacturer’s instructions. The PCR products were purified using Ampure XP beads (A63882; Beckman, Brea, CA, USA). Quantification of the purified samples was performed using Qubit v.2.0 and agarose gel electrophoresis. Library preparation was performed using a Library Construction Kit (MD001T-P4-B; Matridx, Hangzhou, China). Sequencing was performed using the Illumina Netseq 550 platform and quality control was performed under default parameters. Sequences of the A/Anhui/1-DEWH730/2013/H7N9 virus were selected as reference databases, and in addition, the site was considered to be mutated if the frequency of variation was more than 50%.

The NA-Fluor Influenza Neuraminidase Assay Kit (4457091; Applied Biosystems, San Francisco, CA, USA) was used to detect the inhibition of NA enzyme activity (NAI) in the H7N9 virus strains. The IC_50_ value (reduced enzyme activity by 50%) was calculated and compared with the standard resistant strain GD003R294K and standard sensitive strain GD003R294. According to the WHO criteria for determining influenza drug-resistant strains, an increase in the IC_50_ lower than ten-fold compared with the reference virus strain IC_50_ was considered as normal inhibition. When the increase factor was between 10- and 100-fold, the sensitivity was reduced. When the increase was >100-fold, the sensitivity was significantly reduced.

### 2.7. Statistical and Analysis

R294K mutation abundance was calculated using the mean concentration (MT/(MT + WT) × 100%). Repeatability was verified using the CV value. Statistical analyses, which included the paired samples *t*-test, were performed using SPSS v.25.0. Statistical significance was set at *p* < 0.05. The IC_50_ values from the NAI test were calculated using GraphPad Prism 7.

## 3. Results

### 3.1. Determining the Optimal Annealing Temperature

The total number of droplets was >10,000, and adequate separation between negative and positive droplets was observed when the annealing temperature was between 55.3 and 59.0 °C. However, the highest numbers of nucleic acid copies were detected at 58.0 °C and 56.9 °C for MT and WT, respectively (Figure 1). The higher the annealing temperature selected, the greater the guarantee of specificity and sensitivity of PCR amplification. According to the data presented in Figure 1, an annealing temperature of 58.0 °C was chosen for this study. In the figure, both the MT and WT detection results were considered comprehensively.

### 3.2. Evaluating Repeatability, Sensitivity, and Specificity

Repeatability, sensitivity, and specificity were the main indicators used to evaluate the developed method. The CV was <10% when the concentration of the WT and MT plasmids was high and the CV >10% when the R plasmid concentration decreased to 5.30 copies/μL and the K plasmid concentration decreased to 1.12 copies/μL (Table 2).

Probit regression analysis was applied to determine the LOD of the dd PCR and qPCR. Setting 95% as the positive rate, the detection limits of dd PCR were 4.00 (95% CI: 3.05–6.67) copies/μL for WT plasmids and 6.15 (95% CI: 4.59–10.47) copies/μL for MT plasmids; the detection limits of qPCR were 6.13 (95% CI: 4.87–9.13) copies/μL for WT plasmids and 2.49 (95% CI: 1.89–4.46) copies/μL MT plasmids. The paired samples *t*-test showed that the sensitivities of dd PCR and qPCR were not significantly different (*p* = 0.625).

Both dd PCR and qPCR detected the RNA of GD003 R294 and GD003 R294K, as well as that of the other six respiratory virus samples, including H1N1 pdm09, H3N2, B/Victoria, B/Yamagata, H5N1, and H9N2. The results showed that only GD003 R294, GD003 R294K, and the mixture of GD003 R294 and GD003 R294K were positive, while the other influenza viruses were all negative (Figure 2).

### 3.3. Evaluating the Detection Limit of the R294K Mutation

The positive rate gradually decreased to 1.52, 0.63, 0.29, 0.09, and 0.05%, corresponding to the MT plasmid concentrations. When the positive rate accounted for 0.05%, all the plasmids were detected; in addition, the mean concentration of MT plasmids was 6.025 copies/μL (Table 3), which was consistent with the LOD of MT plasmids (6.15 copies/μL).

### 3.4. Using RT-ddPCR to Detect R294K in H7N9 Virus Clinical Samples

Verifying the clinical application of a newly established method is essential. In this study, RT-dd PCR was used to analyze the 89 H7N9 virus clinical samples. Two clinical samples exhibited the R294K mutation, named A/Zhejiang/65-1/2017 (H7N9) and A/Zhejiang/70-2/2017 (H7N9 virus); the concentrations of the mutant gene were 298 copies/reaction and 46 copies/reaction, respectively (Figure 3). The other 87 samples all displayed a WT signal, but no R294K mutant signal. The two samples were analyzed in eight replicates to calculate the rate of mutation, and the results showed that the mutation abundance of A/Zhejiang/65-1/2017 (H7N9 virus) and A/Zhejiang/70-2/2017 (H7N9 virus) were 99.94% and 95.08%, respectively (Table 4).

### 3.5. R294K Mutation in the Genotypic and Phenotypic Analyses

NGS is a reliable method for detecting mutations; therefore, it was used to evaluate the accuracy of resistant site mutations detected by dd PCR. The WT strain A/Zhejiang/YCHH/2016 (H7N9 virus) and MT strains A/Zhejiang/65-1/2017 (H7N9 virus), and A/Zhejiang/70-2/2017 (H7N9 virus) were detected by NGS. The vaccine strain A/Anhui/1-DEWH730/2013 was used as a reference. The base G at position 882 was substituted with A in A/Zhejiang/65-1/2017 (H7N9 virus) and A/Zhejiang/70-2/2017 (H7N9 virus) compared with the reference sequences, which contributed to the substitution of the amino acid at site 294 (R294K), an essential site associated with drug resistance. According to the analysis of variation, the mutation abundance of A/Zhejiang/65-1/2017 (H7N9 virus) and A/Zhejiang/70-2/2017 (H7N9 virus) was 99.94% and 91.35% respectively, and the mutation abundance of A/Zhejiang/YCHH/2016 (H7N9 virus) was 0% (Table 4).

Resistant phenotype analysis is the gold standard for detecting the drug resistance of the influenza virus [11]. A/Zhejiang/YCHH/2016(H7N9 virus), A/Zhejiang/65-1/2017 (H7N9 virus) and A/Zhejiang/70-2/2017 (H7N9 virus) exhibited susceptibility to oseltamivir and zanamivir in the NAI test. The IC_50_ values of A/Zhejiang/65-1/2017 (H7N9 virus) and A/Zhejiang/70-2/2017 (H7N9 virus) were 728.3 nM and >1000 nM for oseltamivir, respectively, which were both >1000-fold higher than that of the standard strain (Table 5). For zanamivir, the IC_50_ values were 18.44 nM and 12.05 nM, respectively, for the above-mentioned strains compared with that of the standard strain, and the inhibition multiple values were 19.83 and 12.96, respectively; therefore, the sensitivity was reduced. The WT isolated A/Zhejiang/YCHH/2016 (H7N9 virus) strain was susceptive to oseltamivir and zanamivir.

## 4. Discussion

Protein mutations can significantly impact viral infectivity, including the H7N9 virus [12]. The R294K mutation at binding site 294 of N9, for instance, can lead to oseltamivir resistance [4]. The drug-resistant H7N9 virus poses a significant challenge to clinical treatments, and traditional NAI testing and sequencing methods to assess drug resistance are both time-consuming and complex. Thus, it is essential to develop a more convenient method for detecting drug resistance mutations at the molecular level. Thanks to its unique advantages, RT-ddPCR has been successfully applied in various clinical fields. For instance, Tatiana F. et al. developed an RT-dd PCR protocol to quantify foot and mouth virus RNA [13], and Mairiang et al. established an RT-dd PCR method for the detection and quantification of dengue virus [14]. Therefore, using RT-ddPCR for the absolute quantification of the R294K mutation in the H7N9 virus avian influenza virus holds great promise for clinical application.

In the current study, an RT-dd PCR protocol was established to detect the R294K drug resistance mutation in the H7N9 virus, and its repeatability, sensitivity, and specificity were evaluated using plasmids. Plasmid-derived DNA standards are widely used in qPCR due to their stability and easy preparation. However, different amplification reagents and conditions are required for DNA or RNA templates in ddPCR, and the effect of reverse transcription on reproducibility and sensitivity has not been considered. Nonetheless, plasmids have been used in several studies to evaluate the sensitivity of RT-ddPCR [14,15]. In our study, there was no significant difference in sensitivity between qPCR and ddPCR (*p* = 0.63). However, both Yan et al. and Whale et al. reported that RT-ddPCR was more sensitive than RT-qPCR [16,17], but their templates were plasmids. Despite the fact that RT-ddPCR is not more sensitive than RT-qPCR, one of its significant advantages is that it does not require a standard curve, allowing for absolute quantification rather than relative quantification by qRT-PCR [18]. As a result, quantification results are highly reproducible, and inter-laboratory variation is minimal [14].

In this study, 2 out of 89 H7N9 virus clinical samples showed the R294K mutation. The mutation abundances for the two strains, A/Zhejiang/65-1/2017 (H7N9 virus) and A/Zhejiang/70-2/2017 (H7N9 virus), were 99.94% and 95.08%, respectively. These two strains were also analyzed using NGS with the same RNA and showed mutation abundances of 99.94% and 91.35%, respectively. The RT-dd PCR and NGS mutation abundances for A/Zhejiang/65-1/2017 (H7N9 virus) were identical; however, the RT-dd PCR mutation abundance for A/Zhejiang/70-2/2017 (H7N9 virus) was higher than that detected by NGS. When the mutation abundance is below 10%, qPCR and Sanger sequencing results are unreliable [19]. Strains show drug sensitivity in conventional NAI methods when the R294K mutation abundance is below 35% [20]. In contrast, dd PCR can detect mutations as low as 0.1% [17]. Our results also showed that RT-dd PCR could detect R294 at proportions as low as 0.06% (Table 4), consistent with the NGS results. However, NGS is more complex than dd PCR in terms of procedure and result interpretation. Therefore, dd PCR can effectively detect rare mutant SNPs and the presence of R294K drug-resistant mutations [21].

The H7N9 virus can have a significant impact on the lives of infected patients and the economy of a country, especially when it involves drug-resistant mutations. It is essential to develop faster and more accurate detection methods for the early diagnosis of these mutations [22]. qPCR cannot perform absolute quantification, NGS is complicated to operate, and NAI tests are time-consuming and less sensitive. In contrast, dd PCR can provide absolute quantification with higher precision and reduced PCR bias. This reduces the influencing factors and provides more reliable and reproducible results compared to traditional methods. Therefore, the RT-dd PCR method developed in this study is suitable for detecting the R294K mutation in the H7N9 virus. Meanwhile, there are still some limitations to the RT-dd PCR method developed in this study. Compared with the advantages of NGS in detecting various types of mutations (including new or unknown drug-resistant mutations) [23,24], RT-dd PCR can only detect specific known mutation sites and cannot discover new or rare mutations [25,26]. In addition, RT-dd PCR also requires larger sample sizes and higher mutation frequency to achieve more reliable results [27]. Therefore, when using RT-dd PCR to detect R294K mutation sites in H7N9 virus strains, other important or potential drug-resistant mutation sites may be missed.

## Figures and Tables

**Figure 1 viruses-15-00983-f001:**
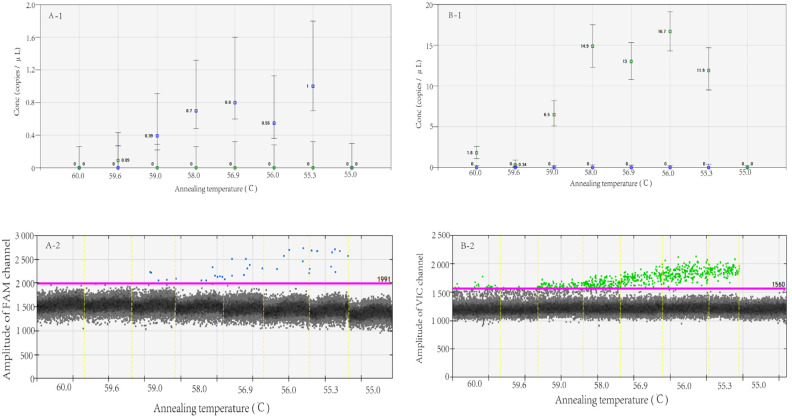
MT (**A**)- and WT (**B**)-type plasmids detected by RT-dd PCR at different annealing temperatures. Note: The *X*-axis of **A-1**, **B-1**, **A-2** and **B-2** indicates different amplification annealing temperatures; the *Y*-axis of **A-1** and **B-1** indicates the number of gene copies per microliter of reaction system for FAM and VIC channels, the *Y*-axis of **A-2** and **B-2** indicates the amplitude of FAM and VIC channels.

**Figure 2 viruses-15-00983-f002:**
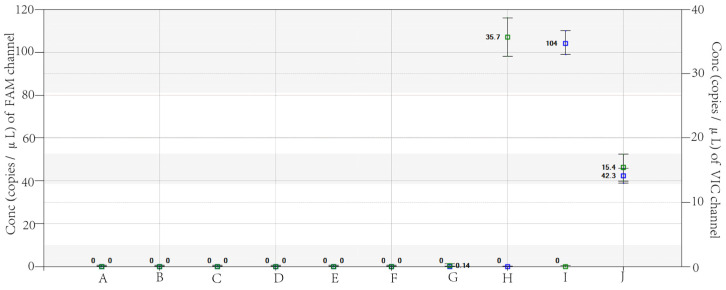
Specific detection results in VIC and FAM channels of dd PCR. Note: The *X*-axis shows different amplification templates, A represents nuclease-free water, while B–J represent the nucleic acid of H1N1 pdm09, H3N2, B/Victoria, B/Yamagata, H5N1, H9N2, GD003 R294, GD003 R294K, and the mixture of GD003 R294K and GD003 R294, respectively. The *Y*-axis represents the number of gene copies per microliter of reaction system for FAM and VIC channels.

**Figure 3 viruses-15-00983-f003:**
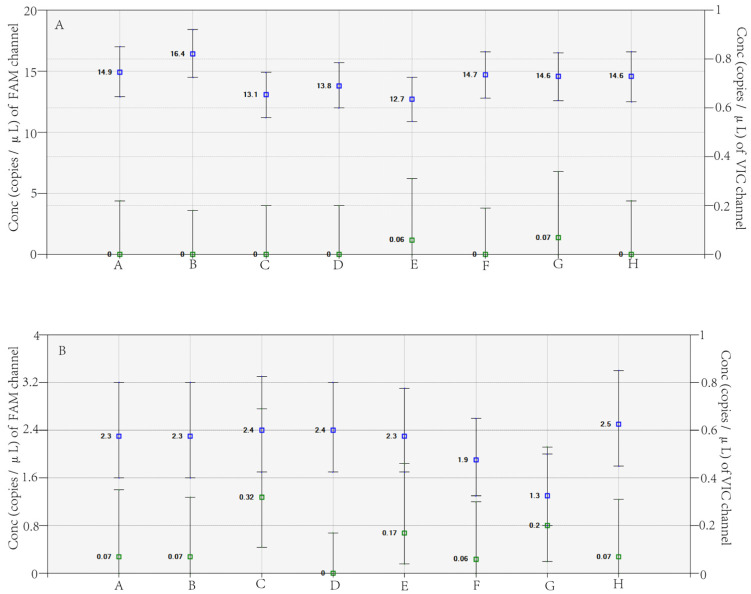
A/Zhejiang/65-1/2017 (H7N9 virus) (**A**) and A/Zhejiang/70-2/2017 (H7N9 virus) (**B**) detected in eight replicates by RT-dd PCR. Note: The *X*-axis represents the results of clinical samples from 8 parallel tests, while the *Y*-axis indicates the number of gene copies per microliter of reaction system for FAM and VIC channels.

**Table 1 viruses-15-00983-t001:** Primer and probe sequences for the detection of the R294K mutation in H7N9 virus.

Primers/Probes	Sequences 5′-3′	Position
R294K-F	TGACTGGAACTGCTAAGCAYATTGA	768–792
R294K-R	TGTCATTGCTACTGGRTCTATCTGA	874–898
294K-Pb1	FAM-CACATGCAAGGACA-MGB	832–845
R294-Pb2	VIC-CACATGCAGGGACA-MGB	832–845

**Table 2 viruses-15-00983-t002:** dd PCR detection repeatability of the R294K mutation in WT and MT plasmids.

Type	Detection Channel	Number	Results (Copies/μL)								Mean (Copies/μL)	SD	CV (%)
WT	VIC	R1	8000	8500	7860	7930	7980	8400	8400	8100	8146.25	248.94	3.06
		R2	639	660	681	684	636	624	667	680	658.88	23.20	3.52
		R3	47.3	47.2	47.2	46.7	48.5	48.3	47.8	47.3	47.54	0.61	1.29
		R4	4.8	4.3	5.7	5.8	5.8	5.8	5.0	5.2	5.30	0.57	10.72
MT	FAM	K1	539	520	540	546	510	523	544	532	532	12.88	2.4
		K2	48.1	49.8	51.1	53.0	52	49.3	42.3	46	48.95	3.48	7.1
		K3	10.8	10.1	11.0	10.3	11.3	11.9	10.8	10.8	10.87	0.66	6.07
		K4	1.0	1.0	1.1	0.96	0.93	1.1	1.8	1.1	1.12	0.28	25.03

Note: Numbers R1–R4 indicate a ten-fold dilution of WT plasmids, R1 was the original plasmids, R2 was the product of a ten-fold dilution of R1, and so on. Numbers K1–K4 indicate a ten-fold dilution of MT plasmids, K1 was the original plasmids, K2 was the product of a ten-fold dilution of R1, and so on.

**Table 3 viruses-15-00983-t003:** dd PCR detection limit of the R294K mutation in WT and MT plasmids.

Number	Detection Channel	Results (Copies/μL)								Mean (Copies/μL)	Mutation Abundance (%)
M1	VIC	8700	8160	9360	8320	8860	8680	9180	9200	8807.6	1.52
	FAM	134	144	96	140	140	138	158	134	135.6	
M2	VIC	9160	8600	9060	8740	8900	8840	9060	9340	8962.5	0.63
	FAM	44	70	64	52	46	60	62	60	57.25	
M3	VIC	8980	9120	8780	8700	9500	9700	9560	9280	9202.5	0.29
	FAM	28	24.8	30.2	15.2	26.4	26.8	22.2	38	26.45	
M4	VIC	10,580	10,460	11,120	10,200	13,280	10,140	10,260	10,820	10,857.5	0.09
	FAM	10	8.6	9.6	10.4	6.8	12.4	8.4	12.4	9.825	
M5	VIC	10,580	10,920	11,140	11,420	11,020	11,180	10,780	11,420	11,057.5	0.05
	FAM	2.8	7.2	3.6	2.6	10.2	5.4	8.6	7.8	6.025	

Note: Numbers M1–M5 indicate the mixture of different contents of WT and MT plasmids.

**Table 4 viruses-15-00983-t004:** The proportion of R294K mutant copies detected by RT-dd PCR and NGS.

Strains	NGS						dd PCR	
	R294			294K			R294	294K
	A	G	G	A	A	G		
A/Zhejiang/65-1/2017(H7N9 virus)		0.06%			99.94%		0.06%	99.94%
A/Zhejiang/70-2/2017(H7N9 virus)		8.65%			91.35%		4.92%	95.08%
A/Zhejiang/YCHH/2016(H7N9 virus)		100%			0		100%	0

**Table 5 viruses-15-00983-t005:** NAI results.

Strains	Oseltamivir		Zanamivir	
	IC_50_ (nM)	Inhibition (Multiple)	IC_50_ (nM)	Inhibition (Multiple)
A/Zhejiang/65-1/2017(H7N9 virus)	728.30	>1000.00	18.44	19.83
A/Zhejiang/70-2/2017(H7N9 virus)	>1000.00	>1000.00	12.05	12.96
A/Zhejiang/YCHH/2016(H7N9 virus)	0.66	0.92	1.05	1.13
GD003 R294	0.72	1.00	0.93	1.00
GD003 R294K	>1000.00	>1000.00	10.75	11.56

## Data Availability

The raw data that support the conclusions of this article will be made available by the authors, without undue reservation.

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
