# Peer review of "Detecting the Neuraminidase R294K Mutation in Avian Influenza A (H7N9) Virus Using Reverse Transcription Droplet Digital PCR Method"

_viruses, 2023, doi:10.3390/v15040983_

Round 1
Reviewer 1 Report
1. An assessment (in two or three sentences) that captures the major conclusions of the review in a concise manner accessible to a wide audience.
Rt-dd PCR can be used as a method to detect the R294K gene mutation in H7N9, the sensitivity is comparable to RT-qPCR, and the accuracy is comparable to NGS. It has the advantages of being able to perform absolute quantification, eliminating the need for a calibration standard curve, and being simpler in both experimental operation and result interpretation than NGS.
2. A public review that details the strengths and weaknesses of the manuscript before you, and discusses whether the authors’ claims and conclusions are justified by their data.
This paper contributes dd PCR scheme for mutation detection, which enables quick and simple detection of clinical sample. And the proposed scheme outperforms the state of the techniques in certain aspects and can monitor for changes in resistance genes. The feasibility of the technology was proved by the data from repeatability, sensitivity and specificity experiments.
3. A set of private recommendations for the authors that outline how you think the science and its presentation could be strengthened.
In introduction,the author took a lot of part to explain background on drug resistance, introduction should provide research gap/research objectives, such as shortcomings of existing technologies, unmet needs and advantages of using this approach. Besides, author should provide a brief explanation of the Rt-dd PCR, and should propose the purpose of the research.
In materials and methods part, whether the sequences targeted by the two probes are opposite, which is inconsistent with the subsequent results; the methods detecting specificity is incomplete.
Figure 1. Confused about why the MT/ blue-marked square indicate the concentration of VIC channel (is this a mistake?), and lack of explanation of coordinates. It seems the copies of positive droplets are all low in (A)? Why is 55℃ set as a negative control, and is it possible that 55 ℃ is the most suitable annealing temperature for MT(A)? In addition, more clear explanation that choose 58℃ as annealing temperature for following experiments is needed.
Table2, the note of number “R1, K1…” could be added. “The CV was < 10% when the concentration of the WT and MT plasmids was > 10 copies/μL”, how can this conclusion be drawn from Table 2?
Table3, the note of number “M1, M2…” could be added. “in addition, the mean concentration of MT plasmids was 6.025 copies/reaction, which was consistent with the LOD of MT plasmids (6.15 copies/reaction)” cannot explain the detection limit of the R294K mutation, may need to add a group which has lower concentration of MT plasmids to come to a conclusion about detection limit.
In discussion, “Our results also showed that RT-dd PCR could detect R294 at proportions as low as 0.06% (Table 3)”, is it referring to Table 4?
Author Response
Thank you for providing feedback on our manuscript. We value your suggestions as they are instrumental in improving the quality of our study. After careful consideration, we have made the necessary revisions and incorporated the feedback. The significant changes made to the manuscript along with our response to your comments are listed below:
Q: A set of private recommendations for the authors that outline how you think the science and its presentation could be strengthened.
- In introduction, the author took a lot of part to explain background on drug resistance, introduction should provide research gap/research objectives, such as shortcomings of existing technologies, unmet needs and advantages of using this approach. Besides, author should provide a brief explanation of the Rt-dd PCR, and should propose the purpose of the research.
Reply: Thank you very much for your suggestion. In the introduction, we have presented an overview of the current state of research and highlighted the benefits of RT-dd PCR. Additionally, we have included the purpose of this study in the final paragraph of the Introduction.
- In materials and methods part, whether the sequences targeted by the two probes are opposite, which is inconsistent with the subsequent results; the methods detecting specificity is incomplete.
Reply: According to the reviewer's suggestions, we have verified the sequences and confirmed their accuracy. Additionally, we have examined the figures in the manuscript and identified that the figure notes contained errors. We apologize for these mistakes and have amended them in the original manuscript. Regarding specificity, the detection of all subtypes of NA is necessary to evaluate the method's specificity. However, due to limited availability of NA types of avian influenza virus in our laboratory, it was challenging for us to perform such an analysis. Nevertheless, the study uses primers and probes that were specifically designed for the NA sequence of H7N9 based on the NCBI and GISAID influenza virus sequence databases. A BLAST search of the designed primers and probes in the NCBI revealed high homology with H7N9. We also assessed the specificity using common influenza viruses, including influenza A viruses (H1N1 and H3N2), influenza B viruses (Victoria and Yamagata), and avian influenza A virus strains (H5N1, H7N9, and H9N2). The results, shown in Figure 2, demonstrate that only H7N9 can be detected among these influenza viruses. Therefore, we conclude that the methods employed in this study can effectively and specifically detect N9.
- Figure 1. Confused about why the MT/blue-marked square indicate the concentration of VIC channel (is this a mistake?), and lack of explanation of coordinates. It seems the copies of positive droplets are all low in (A)? Why is 55℃ set as a negative control, and is it possible that 55 ℃ is the most suitable annealing temperature for MT(A)? In addition, more clear explanation that choose 58℃ as annealing temperature for following experiments is needed.l
Reply: According to the reviewer's suggestion, we have included coordinate explanations in the figure notes. Additionally, after confirming the accuracy of the figures, we discovered errors in the notes. We have modified them to read: "Note: The X-axis represents the different annealing temperatures and the Y-axis represents the detected concentration of nucleic acid. Blue-marked squares indicate the concentration of the FAM channel, while green-marked squares indicate the concentration of the VIC channel" for Figure 1 in the manuscript. Regarding Figure 1, please note that the results represent independent assays to determine the copies of nucleic acid at different annealing temperatures for the MT and WT plasmids. The results for the MT and WT plasmids should not be compared, as they were analyzed separately. It is essential to highlight that different annealing temperatures can impact the sensitivity and specificity of PCR amplification. These effects are influenced by various factors, such as primer length, base composition and concentration, and the length of the target base sequences. Based on the number of nucleotides in the primers, the G+C content, and the Tm value, the primer synthesis company recommended 55℃ as the ideal starting point for selecting the optimal annealing temperature in this study. However, we applied 55℃ as the negative control to assess whether primers would appear as primer dimers. It is noteworthy that we did not use 55℃ as the annealing temperature to amplify WT and MT. A lower annealing temperature could affect specificity, which was a concern in this study. We have revised the sentence: “To ensure the amplification efficiency and specificity of dd PCR, an annealing temperature of 58.0 ℃ was used throughout the remainder of the study.” to "For optimal sensitivity and specificity of PCR amplification, a higher annealing temperature is preferred. Based on a comprehensive analysis of the MT and WT detection results presented in Figure 1, an annealing temperature of 58.0 ℃ was identified to achieve optimal amplification efficiency and specificity. Therefore, this temperature was used throughout the remainder of the study." This revised sentence provides a more comprehensive and detailed explanation of the choice of the annealing temperature used in the study.
- Table2, the note of number “R1, K1…” could be added.“The CV was < 10% when the concentration of the WT and MT plasmids was > 10 copies/μL”, how can this conclusion be drawn from Table 2?
Reply: According to the reviewer’s suggestion, we have added the note of number R1-R4 and K1-K4 under Table 2 in the manuscript. Analysis the CV value in Table 2, it was found that CV < 10% when the plasmid concentration was high, the CV > 10% when the R plasmid concentration decreased to 5.30 copies/μL and the K plasmid concentration decreased to 1.12 copies/μL, the sentence in manuscript was revised to “The CV was < 10% when the concentration of the WT and MT plasmids was high, the CV > 10% when the R plasmid concentration decreased to 5.30 copies/μL and the K plasmid concentration decreased to 1.12 copies/μL (Table 2)”.
- Table3, the note of number “M1, M2…” could be added. “in addition, the mean concentration of MT plasmids was 6.025 copies/reaction, which was consistent with the LOD of MT plasmids (15 copies/reaction)” cannot explain the detection limit of the R294K mutation, may need to add a group which has lower concentration of MT plasmids to come to a conclusion about detection limit.
Reply: According to the reviewer’s suggestion, we have added “Note: Number M1-M5 indicate the mixture of different content of WT and MT plasmids” under the Table 3 in the manuscript. Due to flaws in sample collection, some samples were so low in concentration that they could not be detected
- In discussion, “Our results also showed that RT-dd PCR could detect R294 at proportions as low as 0.06% (Table 3)”, is it referring to Table 4?
Reply: Thank you for the reminder, we have revised the sentence to “Our results also showed that RT-dd PCR could detect R294 at proportions as low as 0.06% (Table 4), consistent with NGS results” in the original manuscript.

Reviewer 2 Report
The authors in the present manuscript describe "Reverse transcription droplet digital PCR detection method for R294K mutation in avian influenza virus H7N9". The authors have optimized the method by using Wild type and mutant plasmids. The more robust method would have been by using RG developed virus creation having the desired mutation.
Author Response
Thank you for providing feedback on our manuscript. The revised and updated manuscript has been uploaded.

Reviewer 3 Report
The manuscript submitted by Xiuyu Lou described a droplet digital PCR (ddPCR) based assay for detecting single nucleotide polymorphism (SNP) at the amino acid position 294 of NA protein of avian influenza A virus. In comparison to conventional qPCR, the ddPCR assay has the advantage in detecting the targets of low abundance and can discriminate SNP in a single assay. Moreover, the ddPCR assay described in this study exhibited high sensitivity, specificity, and reproducibility, and has the potential to be used for clinical sample diagnosis and field surveillance. Although this assay can be useful for the rapid detection of neuraminidase inhibitor resistance mutations during viral infection in human hosts, some sections of the study require clarification. There are many grammar errors that must be corrected. Figure legends miss essential information.
Minor and major comments:
1. Line 1-2: In the title, please indicate “R294K” mutation is in the NA protein. Change “avian influenza virus H7N9” to “avian influenza A(H7N9) virus”. Also throughout the manuscript, try not to use just “H7N9” when “influenza A(H7N9) virus” or “H7N9 virus” should be used instead.
2. Line 58: what does “low load” mean here? Do you mean “detection of low abundance”?
3. Line 80-86: for primer and probe design, please add the nucleotide position information for primers and probes. Meanwhile, please indicate whether primers/probes are N9 specific.
4. Line 98: please specify which type of clinical sample was collected for detection?
5. Line 116” A reader was used to read measure the signals of the PCR products”. Delete second “read” in this sentence.
6. Line 121. What does “copy/reaction=20x copy/ul” mean? Do you mean “copy/reaction in 20 ul of reaction”? Please clarify.
7. Line 130: please specify the range of wt and MT gene copy numbers in serially diluted plasmids.
8. Line 135: “the plasmids were continuously diluted twice”, do you mean “2-fold serial dilution? Here?
9. Line 142: a mixture of wt an MT was prepared. Please indicate the total gene copy number in the mixture.
10. In the “Methods” section, there is no description on RT-qPCR, which was mentioned in the discussion, and need to add a short description on data analysis for NGS and the threshold for variant calling.
11. Figure 1, there is no description on how the plasmids were prepared for this assay. What was the input gene copy number in the reaction? Different copy numbers were detected when different annealing temperatures were used? Which was more consistent with the input gene copy number? Why copy number detected in fig 1a was almost 10-fold lower than that in fig 1b? Was lower input was used in fig 1a. For anneal temperature optimization, it might be better to show scatter plots to visualize the separation of negative and positive droplets. Figure1 legend needs to provide additional information to help readers understand the graph.
12. Fig 2 and 3. Y axis is the concentration in copies per ul. Is it the copy number per ul of extracted RNA or ul of reaction or ul of clinical sample?
13. Line 186: “blue-marked square indicate the … ”. should be “squares”. Same case for figure 2,3 legend.
14. Line 214-219: Although it is justifiable to use the plasmid template when optimizes the conditions (such as the annealing temperature), it is necessary to use serially diluted RNA templates when determines the detect limit.
15. Table 3 is confusing. In table head, results were listed as copies/ul, but in the text, it was mentioned as 6.025 copies/reaction. According to Line 121, “copy/reaction=20x copy/ul”. Please be consistent and clarify the results.
16. Line 283: I cannot find in the text the comparison was made between qPCR and ddPCR.
Author Response
Thank you for providing feedback on our manuscript. We value your suggestions as they are instrumental in improving the quality of our study. After careful consideration, we have made the necessary revisions and incorporated the feedback. The significant changes made to the manuscript along with our response to your comments are listed below:
- Line 1-2: In the title, please indicate “R294K” mutation is in the NA protein. Change “avian influenza virus H7N9” to “avian influenza A(H7N9) virus”. Also throughout the manuscript, try not to use just “H7N9” when “influenza A(H7N9) virus” or “H7N9 virus” should be used instead.
Reply: Thank you very much for your valuable advice. We've made some changes and modified the title to "Detecting the Neuraminidase R294K Mutation in Avian Influenza A (H7N9) Virus Using Reverse Transcription Droplet Digital PCR Method".
- Line 58: what does “low load” mean here? Do you mean “detection of low abundance”?
Reply: Yes, we mean this method can detect low abundance, thank you for your advise, and we have revised the sentence to “RT-dd PCR is known for its high sensitivity, accuracy, and low-abundance detection, making it a popular choice in various fields such as cancer gene and virus detection”.
- Line 80-86: for primer and probe design, please add the nucleotide position information for primers and probes. Meanwhile, please indicate whether primers/probes are N9 specific.
Reply: As per the reviewer's suggestion, we have added the nucleotide position information for primers and probes in Table 1. Regarding the specificity of the primers and probes for N9, detecting all subtypes of NA is required to fully evaluate the method's specificity. However, it was not feasible for us to detect all NA subtypes as there are limited NA types of avian influenza virus available in our laboratory. Nevertheless, the primers and probes utilized in this study were specifically designed for the NA sequence of H7N9, drawing on the NCBI and GISAID influenza virus sequence databases. BLAST analyses conducted in the NCBI database showed that the primers and probes displayed high homology with H7N9. To assess the specificity of the primers and probes, we subjected common influenza viruses to detection, including influenza A viruses (H1N1 and H3N2), influenza B viruses (Victoria and Yamagata), and avian influenza A virus strains (H5N1, H7N9, and H9N2). The results, as shown in Figure 2, demonstrate that only H7N9 was detected among these influenza viruses. As such, we are confident in asserting that the methods employed in this study are capable of specifically detecting N9.
- Line 98: please specify which type of clinical sample was collected for detection?
Reply: According to the reviewer’s suggestion, we add the type of clinical sample, and the sentence was revised to “H7N9 virus RNA was extracted from 200 μL of clinical samples, include swabs, sputum, alveolar lavage fluid, using the Qiagen RNeasy Mini Kit, according to the manufacturer’s instructions”.
- Line 116” A reader was used to read measure the signals of the PCR products”. Delete second “read” in this sentence.
Reply: This was a careless mistake, thank you a lot for pointing it out. This mistake had been solved.
- Line 121. What does “copy/reaction=20x copy/ul” mean? Do you mean “copy/reaction in 20 ul of reaction”? Please clarify.
Reply: Yes, the “copy/reaction=20x copy/ul” means “copy/reaction in 20 ul of reaction”. We have added additional explanations to the manuscript.
- Line 130: please specify the range of wt and MT gene copy numbers in serially diluted plasmids.
Reply: According to the reviewer’s suggestion, we add the original concentration of WT and MT in the manuscript, the sentence has been revised to “A ten-fold gradient was used to dilute the WT and MT plasmids four times, with the original concentration of 8146.25 copies/μL and 532 copies/μL for WT and MT, respectively, dd PCR was used to detect each dilution, and each dilution plasmid was detected in eight repeats”.
- Line 135: “the plasmids were continuously diluted twice”, do you mean “2-fold serial dilution? Here?
Reply: Yes, “the plasmids were continuously diluty twice” means “The plasmids were 2-fold serial dilution continuously” . We have added additional explanations to the manuscript.
- Line 142: a mixture of wt an MT was prepared. Please indicate the total gene copy number in the mixture.
Replay: Thank you for providing valuable feedback. We have made the necessary additions to the corresponding section in the body of the text. The specific content is as follows: The initial concentration of the mixed sample contained both MT plasmid and WT plasmid at a ratio of 1:19 with a total concentration of 10,000 copies/μL. Specifically, WT plasmid accounted for 95% of the mixture while MT plasmid accounted for 5%. Further ratios were tested with WT plasmid accounting for 97.5%, and MT plasmid for 2.5%; WT plasmid accounting for 98.75%, and MT plasmid for 1.25%; WT plasmid accounting for 99.375%, and MT plasmid for 0.625%; and finally, WT plasmid accounting for 99.6875%, and MT plasmid for 0.3125%
- In the “Methods” section, there is no description on RT-qPCR, which was mentioned in the discussion, and need to add a short description on data analysis for NGS and the threshold for variant calling.
Reply: As per the reviewer's suggestion, we have added "The reagents, reaction systems, and reaction procedures used were consistent to ensure comparability of the results between dd PCR and qPCR. The qPCR was performed on the ABI 7500" to section 2.4.3. We have also revised section 2.6 to state: "Sequencing was performed using the Illumina Netxseq 550 platform, and quality control was conducted under default parameters. We selected sequences of A/Anhui/1-DEWH730/2013/H7N9 as reference databases. Furthermore, sites were considered mutated if the sequencing depth at the variant site was over 50%." These modifications provide additional details and clarify the sequencing process.
- Figure 1, there is no description on how the plasmids were prepared for this assay. What was the input gene copy number in the reaction? Different copy numbers were detected when different annealing temperatures were used? Which was more consistent with the input gene copy number? Why copy number detected in fig 1a was almost 10-fold lower than that in fig 1b? Was lower input was used in fig 1a. For anneal temperature optimization, it might be better to show scatter plots to visualize the separation of negative and positive droplets. Figure1 legend needs to provide additional information to help readers understand the graph.
Reply: The manuscript has been revised to incorporate the reviewer's suggestions, as shown below: Following the reviewer's recommendation, we have included scatter plots in Figure 1 and provided additional information to enhance comprehension of the data. The results illustrated in Figure 1 were attained from independent assays to assess the copies of nucleic acid for both the MT and WT plasmids, respectively, at different annealing temperatures. It is important to note that the results from the MT and WT plasmids should not be compared due to variations in the experimental approach. In section 2.3, we made revisions regarding the MT and WT plasmids, stating that the recombinant plasmids of the wild type (WT) and mutant type (MT) of the NA gene were synthesized by Shanghai Sangon Company, and were used directly without the extraction of nucleic acid. Once we obtained the plasmids, we divided them to ensure an equal concentration of plasmids was used in the experiment. According to the synthesis report, the initial concentrations of the MT and WT plasmids were 1 and 2, respectively.
- Fig 2 and 3. Y axis is the concentration in copies per ul. Is it the copy number per ul of extracted RNA or ul of reaction or ul of clinical sample?
Based on the reviewer's feedback, we have incorporated a note for the figure. For Figure 2, we have revised the note as follows: "Note: The X-axis shows different amplification templates, A represents nuclease-free water, while B-J represent nucleic acid of H1N1 pdm09, H3N2, B/Victoria, B/Yamagata, H5N1, H9N2, GD003 R294, GD003 R294K, and the mixture of GD003 R294K and GD003 R294. The Y-axis represents the number of gene copies per microliter of reaction system for FAM and VIC channels. The blue-marked square indicates the FAM channel concentration, while the green-marked square indicates the VIC channel concentration." For Figure 3, we have revised the note as follows: "Note: The X-axis represents the results of clinical samples from 8 parallel tests, while the Y-axis indicates the number of gene copies per microliter of reaction system for FAM and VIC channels. The blue-marked square indicates the FAM channel concentration, while the green-marked square indicates the VIC channel concentration."
- Line 186: “blue-marked square indicate the … ”. should be “squares”. Same case for figure 2,3 legend.
Reply: This was a careless mistake, thank you a lot for pointing it out. This mistake had been solved.
14. Line 214-219: Although it is justifiable to use the plasmid template when optimizes the conditions (such as the annealing temperature), it is necessary to use serially diluted RNA templates when determines the detect limit.
Reply: The H7N9 avian influenza virus is an RNA virus, and it is preferable to use an RNA template to determine the limit of detection. But in the design of this study, we refered to references 13-16 in the manuscript, which indicated that plasmids can also be used to determine the limit of detection for RNA viruses, and we also noted this point in the Discussion.
- Table 3 is confusing. In table head, results were listed as copies/ul, but in the text, it was mentioned as 6.025 copies/reaction. According to Line 121, “copy/reaction=20x copy/ul”. Please be consistent and clarify the results.
Reply: According to the reviewer’s suggestion, we re-checked the contents of the manuscript and found that the units in Table 3 had been written incorrectly and that we had revised “copies/reaction” to “copies/μL”.
- Line 283: I cannot find in the text the comparison was made between qPCR and ddPCR
Reply: Thank you for your suggestion. We've added relevant content comparing qPCR and ddPCR in Line 379-384 (Yellow highlights).

Round 2
Reviewer 3 Report
I have reviewed the revised manuscript and I believe that authors have fully addressed my concerns. However, there is one error in the author's reply, which needs to be rectified. In comment 10: "Furthermore, sites were considered mutated if the sequencing depth at the variant site was over 50%." Sequencing depth should be a number, but not a percentage.
Author Response
Thank you for providing valuable feedback. We revised “Furthermore, sites were considered mutated if the sequencing depth at the variant site was over 50%” to “Furthermore, sites were considered mutated if the frequency of variation was over 50%” in the manuscript.
